# 3D-Printed Pectin/Carboxymethyl Cellulose/ZnO Bio-Inks: Comparative Analysis with the Solution Casting Method

**DOI:** 10.3390/polym14214711

**Published:** 2022-11-03

**Authors:** Yeon Ho Kim, Ruchir Priyadarshi, Jin-Wook Kim, Jangwhan Kim, Denis G. Alekseev, Jong-Whan Rhim

**Affiliations:** 1Department of Food and Nutrition, Kyung Hee University, 26 Kyungheedae-ro, Dongdaemun-gu, Seoul 02447, Korea; 2RokitHealth Care Ltd., 9, Digital-ro 10-gil, Geumcheon-gu, Seoul 08514, Korea; 3Samara State Medical University, Ulitsa Artsybushevskaya, 171, Samara 443001, Russia

**Keywords:** 3D-bioprinting, solution casting, bio-inks, antimicrobial activity, cytotoxicity

## Abstract

Bio-inks consisting of pectin (Pec), carboxymethyl cellulose (CMC), and ZnO nanoparticles (ZnO) were used to prepare films by solution casting and 3D-printing methods. Field emission scanning electron microscopy (FE-SEM) was conducted to observe that the surface of samples made by 3D bioprinter was denser and more compact than the solution cast samples. In addition, Pec/CMC/ZnO made by 3D-bioprinter (Pec/CMC/ZnO-3D) revealed enhanced water vapor barrier, hydrophobicity, and mechanical properties. Pec/CMC/ZnO-3D also showed strong antimicrobial activity within 12 h against *S. aureus* and *E. coli* O157: H7 bacterial strains compared to the solution cast films. Further, the nanocomposite bio-inks used for 3D printing did not show cytotoxicity towards normal human dermal fibroblast (NDFB) cells but enhanced the fibroblast proliferation with increasing exposure concentration of the sample. The study provided two important inferences. Firstly, the 3D bioprinting method can be an alternative, better, and more practical method for fabricating biopolymer film instead of solution casting, which is the main finding of this work defining its novelty. Secondly, the Pec/CMC/ZnO can potentially be used as 3D bio-inks to fabricate functional films or scaffolds and biomedical applications.

## 1. Introduction

With the increasing environmental awareness in recent years, the research on biopolymers as sustainable alternatives to synthetic plastics has been exponentially growing. Researchers have been concentrating on using natural/nature-derived polymers as sustainable materials, particularly in food packaging and biomedical engineering applications over the past few years. Several biopolymers, such as proteins, polysaccharides, lipids, and their mixtures, have been investigated in this regard [1]. In addition to being environmentally friendly, biopolymers have enormous promise for delivering functional ingredients and regulating their release, opening the door to the fabrication of active packaging [2]. Additionally, biopolymers are crucial for various advanced applications such as biomedical engineering due to their controlled-release property, non-toxicity, hydrophilicity, biocompatibility, and biodegradability [3]. Despite a tremendous development in biopolymer-related materials and technologies on the lab scale, very few commercial products are available in the market. The most common reason for this is the unavailability of efficient biopolymer processing methods, i.e., biopolymers cannot be processed using conventional polymer production methods such as injection molding, extrusion, and blow molding.

Recently, 3D printing has emerged as a commercial fabrication technique, especially for biopolymers used prominently for biomedical applications. Hull first illustrated 3D printing in 1986 [4]. Recently, this idea has evolved from two-dimensional (2D) printing to 3D shapes, which is realized by building successive layers of materials in the third dimension [5,6]. The 3D printing technology enables rapid industrial prototyping and manufacturing, creating personalized consumer products [7]. Several methods for 3D printing have been used, including selective laser sintering (SLS) [8], stereo lithography apparatus (SLA) [9], fused deposition modeling (FDM) [10], selective laser melting (SLM) [11], and electronic beam melting (EBM) [12]. Technically, 3D printers rely on high temperatures for melting and extruding materials such as poly(lactic-co-glycolic) acid (PLG), poly-lactic acid (PLA), and polycaprolactone (PCL), etc. However, this technology can be applied to only polymers with high melting temperatures, limiting its application areas.

Therefore, 3D bioprinters have been developed to expand the applications of 3D printing in various fields such as biology, medicine, pharmacology, and food science. 3D bioprinters enable the printing of low melting point biological materials. Furthermore, the aqueous-based scaffolds can be directly printed [13]. Such low melting point polymers or aqueous-based biopolymers, with or without functional additives or cells (in the case of biomedical materials) that can be printed via a 3D bioprinter, are termed bio-inks. Several studies have used hydrogel biopolymers such as gelatin, alginate, and agarose as bio-inks [14,15] because these polymers show better interaction with cells and biological surfaces and their viscosities can be easily tuned by regulating temperature. For instance, solutions of some biopolymers, such as agarose, solidify at lower temperatures. This technique has been widely and successfully applied to various biomedical research fields such as regenerative medicine, tissue engineering [16], drug delivery systems, etc., where 3D bioprinted tissues have been used [17]. In addition to biomedicals, food science is another research area where this technology has recently been used for applications such as the fabrication of cultured meat [18].

Another application in food science research where the 3D bioprinting technology can be extremely useful is the fabrication of biopolymeric food packaging films or hydrogels. So far, solution casting has been the most suitable method for fabricating biopolymer films and hydrogels [19,20]. However, this method is difficult to be replicated at an industrial level due to longer drying time and various manual processes. 3D bioprinting can be an alternative to the solution casting method being swift and industrially scalable. Furthermore, due to automatic control over the printing process, the chances of deviation in properties of the fabricated hydrogels and films are much reduced. Although 3D bioprinting technology offers several advantages over conventional lab-scale solution casting methods, there are barely any studies conducted globally to verify its potential in this area.

Recent research in biopolymer-based food packaging films focuses on developing active films with better physicochemical and functional properties. For this purpose, multiple components are mixed to obtain the final formulation having desired properties. For instance, several biopolymers are blended to obtain the desired mechanical attributes, while additives such as nanoparticles are incorporated in the formulation to provide functionalities such as antimicrobial activity. Therefore, this study aims at formulating a bio-ink combining pectin with sodium carboxymethyl cellulose (CMC) incorporated with zinc oxide nanoparticles (ZnONP). Both CMC and pectin are natural plant-based polysaccharides that are renewable biopolymers suitable for fabricating sustainable food packaging films. Further, they have been widely used for fabricating films for this purpose on the lab scale and display good physicochemical properties and potential to be scaled-up on the industrial level. 3D bioprinting technology was employed to fabricate films using the bio-ink formulation, and the physicochemical, functional, and biological properties of the 3D printed films were compared with that of the films prepared by the conventional solution casting method.

## 2. Materials and Methods

### 2.1. Materials

Pectin and zinc chloride were purchased from ES Food Co., Ltd. (Gunpo, Korea). Sodium carboxymethyl cellulose was purchased from Junsei Chemical Co., Ltd. (Tokyo, Japan). Calcium chloride was procured from Sigma-Aldrich (St. Louis, MO, USA). Glycerol and sodium hydroxide were obtained from Daejung chemicals & Metals Co., Ltd. (Siheung, Korea).

### 2.2. Preparation of Zinc Oxide Nanoparticles

Zinc oxide nanoparticles (ZnONP) were prepared using sodium hydroxide and zinc chloride following the previously reported method [21]. For the production of ZnONP, 13.6 g of zinc chloride was mixed while stirring in 960 mL of distilled water at 50–60 °C temperature. After that, 40 mL of 5 M sodium hydroxide solution was continuously added dropwise to the zinc chloride solution, and the reaction was continued for 2 h. The ZnONP, formed as white precipitates, were collected using centrifugation, washed with distilled water (three times) and ethanol (two times), and dried in an oven for 6 h at 70 °C to obtain powdered ZnONP.

### 2.3. Preparation of Pectin/CMC/ZnO Bio-Inks and Films

Pectin/CMC-based bio-inks were formulated using the solution casting method [19,20] and 3D bioprinter. The formulation of the bio-inks is shown in Table 1. The aqueous dispersion of glycerin and different concentration of ZnONP (0 and 3 wt% of Pec/CMC) was prepared in distilled water using an ultrasonicator (FS140 Ultra Cleaner, Fisher Scientific, Pittsburg, PA, USA) for 30 min. When glycerin and ZnONP were dispersed well, CMC and Pectin were added to the aqueous dispersion and dissolved using a mechanical stirrer (PL-FS701, Punglim Tech Co., Ltd., Seoul, Korea) for 1 h at maximum rpm. The calcium chloride was separately dissolved in 10 mL of distilled water and mixed with the above aqueous dispersion, followed by 30 min stirring. The air bubbles in the aqueous dispersion were removed using a vacuum pump for 30 min. Two bio-ink formulations were obtained: (a) Pectin CMC blend without ZnONPs, and (b) Pectin CMC blends added with ZnONPs (3 wt%), designated as Pec/CMC and Pec/CMC/ZnO, respectively. Both the bio-ink formulations were used for film fabrication using solution casting and 3D bioprinting methods.

The prepared bio-ink formulations’ 3D printing (DR. INVIVO 4D2, ROKIT Health Care Co., Ltd., Seoul, Korea) was performed using the printing conditions mentioned in Table 2, aided by the NewCreatorK program (ROKIT Health Care Co., Ltd., Seoul, Korea). The obtained films were denoted as Pec/CMC-3D and Pec/CMC/ZnO-3D.

For the solution casting method, the viscosity of the aqueous bio-inks was first reduced by diluting using an equal volume of distilled water, followed by stirring for proper mixing. The solution was then cast onto a glass plate coated with a Teflon sheet (24 cm × 30 cm) and allowed to dry. The obtained films were denoted as Pec/CMC-SC and Pec/CMC/ZnO-SC.

The films prepared by both methods were then peeled off and preconditioned at 50% RH and 25 °C for 48 h before testing and characterization.

### 2.4. Morphology and Structural Analysis

The microstructure of the composite films was recorded using a field emission scanning electron microscope (FE-SEM, S-4800, Hitachi Co., Ltd., Matsuda, Japan). For analysis, a small piece of the film was mounted on a holder using carbon tape. After that, the samples were sputter coated with platinum and observed with an acceleration voltage of 2 kV.

Spectroscopic methods were employed to determine the chemical structure of the films. Fourier transform infrared (FT-IR) spectra of the films were recorded using an FT-IR spectrometer (TENSOR 37 spectrophotometer with OPUS 6.0 software, Billerica, MA, USA) with an average resolution of 4 cm^−1^. The UV-visible absorption spectra were also obtained using a UV-visible spectrophotometer (Mecasys Optizen POP Series UV/Vis, Seoul, Korea) in the 200–700 nm wavelength range.

### 2.5. Optical Analysis and Surface Color

The light transmittance of the films was analyzed using a UV-visible spectrophotometer (Mecasys Optizen POP Series UV/Vis, Seoul, Korea) in the 200–700 nm wavelength range. The transmittance percentages at 280 nm (T280) and 660 nm (T660) were determined for assessing the UV barrier property and transparency of the films, respectively [22]. 

The surface color of the films was estimated by a chromameter (Minolta, CR-200, Tokyo, Japan). For calibration, a white color plate (*L** = 97.75, *a** = −0.49, and *b** = 1.96) was used, and the color parameters (*L**, *a**, and *b**) were obtained as the mean of the values recorded at five different positions on each sample. Total color difference (Δ*E*) was calculated as follows:(1)$$\Delta E=\sqrt{\left[{\left(\Delta L*\right)}^{2}+{\left(\Delta a*\right)}^{2}+{\left(\Delta b*\right)}^{2}\right]}$$
where Δ*L**, Δ*a**, and Δ*b** are differences in the values between the white plate and the film specimen.

### 2.6. Water Vapor Permeability (WVP) and Water Contact Angle (WCA)

The water vapor permeability of the films was evaluated following the ASTM E96-95 standard method [19,20]. The film sections with size 7.5 cm × 7.5 cm were covered atop the measuring cups containing 18 mL of distilled water, which were stored in the humidity chamber at 25 °C and 50% RH. After that, the weight of each cup was estimated every hour for 8 h. The following equation assessed the WVP (g·m/m^2^·Pa·s) of the samples:(2)$$WVP=\frac{\left(WVTR\;\times \;L\right)}{\Delta p}$$
where *WVTR* is the value of water vapor transmission rate (g/m^2^·s) through a sample, *L* is the mean value of sample thickness (m), and Δ*p* is the partial water vapor pressure difference (Pa) across the two sides of the sample.

The surface hydrophilicity/hydrophobicity of the films was determined by assessing the water contact angle using a WCA analyzer (Phoneix 150, Surface Electro Optics Co., Ltd., Kunpo, Korea). Each sample sized as 3 cm × 10 cm was fixed on the Teflon-coated steel plate (7 cm × 11 cm). After this, 10 µL of distilled water was dropped on the surface of the film sample using a microsyringe. The contact angle was analyzed on both sides of the water drop.

### 2.7. Thickness and Mechanical Properties

The thickness of the films was recorded using a digital micrometer (Digimatic Micrometer, QuantuMike IP65, Mitutoyo, Japan). The thickness measurements were performed at least six random sites on each film, and the values were represented as average and standard deviation.

The tensile properties of the films were estimated by an Instron Universal Testing Machine (Model 5565, Instron Engineering Corporation, Canton, MA, USA). Tensile strength (TS), elastic modulus (EM), and elongation at break (EB) of the films were assessed following the standard test method ASTM D882-88. Sections of each film (2.54 cm × 15 cm) were cut using a precision double cutter (model LB.02/A, Metrotec, S.A., San Sebastian, Spain). After that, the samples were strained under conditions of an initial grip separation of 50 mm and a crosshead speed of 50 mm/min.

### 2.8. Antimicrobial Activity

The antimicrobial activity of the film samples was evaluated against *Staphylococcus aureus* (ATCC 13565, enterotoxin A) and *Escherichia coli* O157: H7 (ATCC 11234) obtained from the Korea Microbial Culture Center (KCCM, Seoul, Korea) and stored in trypsin soy broth (TSB) mixing 20% glycerol at −80 °C. The bacterial cultures were thawed at room temperature just before commencing the test. 100 mL of bacteria were incubated in 10 mL of sterile TSB, followed by incubation at 35 °C for 24 h while shaking at 140 rpm on a rotary shaker (VS-8480SR, Vision, Seoul, Korea). The time-kill assay method was conducted using a reported method with slight modification [19]. For diluting the cultures to ~5–6 log CFU/mL of bacteria, 1 mL of culture solution was mixed with 9 mL of 0.1% sterile peptone water using serial dilution. Next, 2 mL of diluted microbial solution was inoculated in 18 mL of TSB liquid media into an Erlenmeyer flask. Afterward, 100 mg of each film sample was added to the liquid media in the designated flask and incubated at 36 °C for 12 h on a rotary shaker. The media without any film sample was set as a control. The media from each flask was inoculated on TSA plates after regular 3 h intervals (3, 6, 9, and 12 h). The TSA were incubated at 36 °C for 24 h, and viable colonies were counted.

### 2.9. Cytotoxicity

The cytotoxicity assay was conducted using WST (Water-Soluble Tetrazolium salt) EZ-Cytox (Dogenbio, Seoul, Korea). 5000 normal human dermal fibroblast (NDFB) cells were seeded in each well of a 96-well plate 24 h before the treatment in MEM media (Gibco, Waltham, MA, USA) without fetal bovine serum for cell cycle arrest. Next, the samples were treated and cultured for 24 h at 37 °C temperature and 5% CO_2_ concentration in a cell culture chamber. After 24 h, the Ez-Cytox solution was added to each well. After 2 h, the plates were transferred to a microplate reader and detected at 450 nm absorbance (reference 600 nm). Finally, the raw data were analyzed using GraphPad Prism 7.0 (GraphPad, San Diego, CA, USA). 

### 2.10. Statistical Analysis

All tests were performed in triplicates, and the results were reported as mean ± SD (standard deviation). One-way analysis of variance (ANOVA) was used for the presentation of the significant difference of each mean value (*p* < 0.05) by Duncan’s multiple range tests in the SPSS statistical analysis program (SPSS Inc., Chicago, IL, USA).

## 3. Results and Discussion

### 3.1. Morphology, FT-IR, and UV-Vis Spectrum

The morphology of fabricated films was observed using the FE-SEM, and the images are shown in Figure 1. The Pec/CMC-SC film surface was a continuous and compact structure indicating good compatibility between pectin and CMC. No voids and cracks were observed on the surface. However, undulations were prevalent on the surface, presumed to be a result of moisture absorbed from the environment by the hydrogel films. The Pec/CMC-3D films also showed similar morphology but a relatively smooth and plain surface without undulations, indicating less moisture absorption by the films. A probable reason for this can be a relatively better interchain H-bonding interaction between the hydroxyl groups of the polymers in the 3D printed films due to less amount of water in the initial formulation. The solution cast films may have weaker interchain H-bonding interaction due to higher water content in the initial formulation. It is presumed that when the films dry, the evaporated water leaves voids in the films, offering more free -OH groups to interact with the moisture from the environment. Water acts as a natural plasticizer for biopolymer films [23], and it has been reported that increased hydrophilic plasticizer content can increase moisture absorption by the biopolymer films [23]. Further, the ZnONPs added films show similar morphology to the base polymer, and both Pec/CMC/ZnO-3D and Pec/CMC/ZnO-SC films exhibit the presence of ZnONPs in the polymer matrix.

The samples were analyzed using FT-IR and UV-Vis spectrophotometry to investigate the structural and chemical characterization of the Pec/CMC-based films. The FT-IR spectra of all the samples exhibit the characteristic absorption banding pattern of Pec/CMC (Figure 2a), without any significant difference among the samples. The broad bands around 3600–3200 cm^−1^ were due to the O-H stretching vibrations and the intermolecular and intramolecular H-bonding in the polymer chains [24]. The C-H stretching bands from the cellulose structure resulted in the vibration at 2880 cm^−1^ [25]. The absorption peak at 1749 cm^−1^ corresponded to the methyl ester’s stretching vibration in the pectin’s carbonyl group [26]. The sharp bands at 1590 cm^−1^ and 1430 cm^−1^ were attributed to the asymmetric and symmetric structure of the -COO^-^ groups, respectively [27]. In addition, the small absorption peak at 1321 cm^−1^ appeared due to the symmetrical deformations of the CH_2_ associated with the carboxyl groups in the polymers [28]. The FT-IR patterns showed the chemical interaction between pectin and CMC, while the peak positions of these two polymers did not change after mixing with ZnONP, which indicated that ZnONP did not affect the chemical interaction within these two polymers. Furthermore, the results of FT-IR indicate that the interaction within the polymers and the chemical structure of the films did not change either based on the fabrication methods or on the incorporation of ZnONPs.

The UV-vis spectra of Pec/CMC-based films within the wavelength range 200–700 nm are shown in Figure 2b. The absorbance of all samples exhibited the signature peak at 280 nm corresponding to the absorption of UV-B radiation, a characteristic of pectin present in the polymer blend. Beyond 280 nm, no other peak was observed throughout the spectral range, indicating high visible light transparency of the neat films. Further, there was no major difference between the UV-visible spectra of neat Pec/CMC films fabricated by solution casting and 3D bioprinting. However, the spectral patterns of Pec/CMC/ZnONP films displayed a significant difference based on the fabrication method. The Pec/CMC/ZnO-SC film displayed a significantly higher light absorption than the Pec/CMC/ZnO-3D films, indicating a higher opacity of the solution cast films. The probable reason for this may be the fabrication of films in the raster scan mode by the 3D printer, resulting in an ordered assembly of polymer chains and ZnONPs in the matrix [29]. The systematic and uniform arrangement of particularly ZnONPs resulted in enough vacant space in the matrix for the light to pass through. On the other hand, in the case of solution cast films, the polymer chains, and nanoparticles are presumed to be stacked one over the other in the form of a jumbled mesh, thereby blocking a higher amount of incident radiation. Further, a significant absorption peak at 376 nm was obtained for Pec/CMC/ZnO-3D films which is a characteristic of ZnONPs and corresponds to the band gap energy of the semiconductor nanoparticles [30]. This further confirms the ordered arrangement of ZnONPs in 3D printed films. Similar absorption peaks for ZnONPs within the range of 360–370 nm were also observed previously [24,31,32].

### 3.2. Color and Mechanical Properties

The color properties of Pec/CMC-based films are shown in Table 3. The properties of the neat Pec/CMC were not considerably different based on the fabrication method. However, the statistical difference was significant. The Pec/CMC-SC film was slightly brighter, with an L*-value of 90.07 compared to 89.67 for Pec/CMC-3D. Although the yellowness (b* values) was similar for both samples, the redness (a*-value) of the solution cast film was slightly lower than the 3D printed film. The slight variation in the values also resulted in a minute difference between the ∆E values that could not be considered noticeable. Further, as expected, the addition of ZnONPs increased the yellowness of both samples, which may be due to the change in the color of ZnONPs from white to yellow on drying [33]. Interestingly, the change in b*-value was higher in Pec/CMC/ZnO-SC films showing a value of 9.67, compared to Pec/CMC/ZnO-3D films exhibiting a value of around 4.12, indicating higher yellowness of the solution cast films. In addition, the redness decreased for both the films, the reduction being higher for the 3D printed films compared to a slight reduction in solution cast ones. This resulted in a greater ∆E value for the solution cast films than 3D printed films on ZnONP incorporation. Hence, the 3D printing method resulted in a lesser variation in the color properties of the fabricated nanocomposite films, which is an important attribute from the commercial perspective.

The mechanical properties of biopolymer films are important not only for packaging films but also for biomedical applications. The mechanical properties, such as the tensile strength (TS) and elongation at break (EB) of films, were estimated, and the results are exhibited in Table 3. The Pec/CMC-SC films displayed TS and EB values of 12.8 MPa and 27.9%, respectively, indicating their low load-bearing capabilities but high flexibility. On the incorporation of ZnONPs, as expected, the TS value increased by ~88% to reach 24.1 MPa without a statistically significant effect on the E.B. Such increase in the TS of biopolymer films has been widely reported by various researchers [21,28] and is attributed to the strain induced alignment of ZnONPs in the biopolymer matrix, resulting in better load distribution [30].

On the other hand, interesting results were obtained for the Pec/CMC 3D printed films which displayed a much higher TS value of 27.6 MPa, which is higher than not only the neat but also the nanocomposite films produced via the solution casting method. Further, the TS value of the 3D printed films increased to 41.3 MPa on incorporating ZnONPs, indicating an almost 50% increase over Pec/CMC-3D films. In addition, compared to Pec/CMC/ZnO-SC films, this value was ~71% higher. It is noticeable that the EB values of the 3D printed films were much lower than the solution cast films; they were still around 20% for both 3D printed films, indicating decent flexibility. The better mechanical properties of the 3D printed films were due to the ordered arrangement of the components in the film matrix due to the raster mode of film fabrication in which the 3D printer works [29]. In addition, the difference in the water content in the two different formulations are also responsible for this difference in mechanical properties (Appendix A).

### 3.3. Optical Properties

The optical properties were studied to determine the UV barrier and transparency of the films, which are important attributes that help prevent oxidation, photocatalytic reaction, cell damage, and extending the shelf-life of food and medicine. The light transmittance spectra of Pec/CMC-based films are shown in Figure 3. The transmittance patterns of Pec/CMC-3D were almost similar to Pec/CMC-SC, indicating an insignificant difference in the films’ optical properties based on the fabrication method. Both the neat films were highly transparent, especially in the visible region of the spectra showing above 80% light transmission at 600 nm. Furthermore, the transmittance at 280 nm was around 20% for both the films indicating their high UV-barrier effect. This is due to the presence of pectin in the blend structure, as already explained earlier. Further, both films incorporated with ZnONPs exhibited significantly lower transmittance. The Pec/CMC/ZnO-3D film displayed a similar UV-blocking effect (T280 = 22%) to the neat film. However, the transparency in the visible region was significantly lower (T600 = 70%). Whereas, in the case of Pec/CMC/ZnO-SC films, the transparency was greatly reduced (T600 = 27%) due to the jumbled and random arrangement of polymer chains and the ZnONPs in the matrix, as already discussed in previous sections. This decrease in transmission of visible light was attributed to the blocking of light by the ZnONPs in the film matrix, which is in accordance with the phenomenon reported earlier for nanocomposite films, where ZnONPs were added to CMC [24], CMC/starch [34], and pectin [35]. Nevertheless, the UV-blocking property was the highest among all the films (T280 = 4%). Overall, the 3D-printed nanocomposite films displayed better optical properties and were suitable for practical applications.

### 3.4. Water Vapor Barrier Property (WVP) and Water Contact Angle (WCA)

The water vapor permeability (WVP) estimates the amount of water vapor or moisture that can pass through a polymeric film, an important material property to be considered for packaging films and biomedical applications. The results of WVP of Pec/CMC-based films are shown in Figure 4a. Interestingly, the Pec/CMC film fabricated by 3D printing displayed almost 38% lower WVP than the solution cast film. The reason may be the 3D printed films’ decreased interaction with water vapor due to less available free hydroxyl groups [36], as indicated by the SEM data. In addition, for films fabricated by both fabrication methods, neat films displayed a higher WVP value which decreased with the incorporation ZnO nanoparticles. This phenomenon is common for nanocomposite biopolymer films because nanoparticles can occupy vacant space in the polymer matrix and block the path for water vapor to pass through the polymer structure [37]. On the incorporation of ZnONPs, the WVP of solution cast film dropped significantly (*p* < 0.05) from 2.46 × 10^−9^ g·m/m^2^·Pa·s to 2.16 × 10^−9^ g·m/m^2^·Pa·s exhibiting around 14% decrease. The 3D printed nanocomposite film also exhibited a nearly similar reduction in WVP, i.e., by 16% on ZnONP addition. However, the WVP value displayed by these films (1.29 × 10^−9^ g·m/m^2^·Pa·s) was almost 40% lower than the nanocomposite films fabricated via solution casting. These extraordinary results further endorse the high commercial and practical potential of 3D printed biopolymer films.

The water contact angle (WCA) results of the Pec/CMC-based films are displayed in Figure 4b. Although the Pec/CMC-3D film has a slightly higher WCA value, it was not significantly higher than the Pec/CMC-SC film. In addition, there was no significant difference between the WCA values of Pec/CMC-SC and Pec/CMC/ZnO-SC. However, the Pec/CMC/ZnO-3D film represented the least hydrophilic one, exhibiting the highest WCA value of 72.6°, which was significantly (*p* < 0.05) higher than other counterparts. For biopolymer films, as a standard criterion, the materials with WCA < 65° are considered hydrophilic, while those with WCA > 65° are considered hydrophobic [38]. Hence, the Pec/CMC/ZnO-3D film could be considered decently hydrophobic with potential for food packaging applications.

Several plausible synergistic reasons exist for this behavior of the 3D-printed nanocomposite films. As already discussed, the 3D printed film has a high degree of orientation due to the raster fabrication mode employed by this technique. It has been widely reported that a material’s surface energy depends on the surface molecules’ orientation; the higher the orientation, the lower the surface energy, resulting in lower hydrophilicity [39,40]. Furthermore, the Pec/CMC/ZnO-3D film may have the hydrophilic –OH groups oriented towards the interior of the polymer matrix, thereby reducing the surface hydrophilicity. Further, it is presumed that the remaining surface hydrophilic groups, being highly oriented, was occupied in the H-bonding with ZnONPs, thereby making the nanocomposite film even more hydrophilic compared to the near 3D printed film [30].

### 3.5. Antimicrobial Activity

The antimicrobial activity of biopolymer nanocomposite films is one of the important functionalities which can be applied to various research areas such as scaffolds for 3D cell culture, alternative protein foods, edible coating for foods and drugs, functional hydrogel patches, and functional food packaging. Studies to determine the antimicrobial activity of Pec/CMC-based films against *S. aureus* and *E. coli* O157: H7 were conducted using a time-kill assay, and the results are shown in Figure 5. The neat Pec/CMC films fabricated by solution casting and 3D printing displayed similar trends. No antibacterial activity was observed for the films against the exposed bacterial cultures, and the growth pattern of the microbes almost traced that of the control cultures. The Pec/CMC/ZnO films showed strong antimicrobial activity against both bacteria due to the antimicrobial action of ZnONPs. Further, the antimicrobial effect was stronger against Gram-negative *E. coli* bacteria than Gram-positive *S. aureus*, a characteristic of ZnONPs reported by other researchers [41]. Interestingly, among both the fabricated nanocomposite films, the Pec/CMC/ZnO-3D displayed a higher antimicrobial activity against both bacterial strains compared to Pec/CMC/ZnO-SC. A 100% elimination of *S. aureus* colonies was observed within 6 h for 3D printed film, whereas solution cast films exhibited a similar effect after 9 h. Similarly, the complete eradication of *E. coli* occurred within 3 h when exposed to 3D printed films, compared to 6 h on exposure to solution cast films. The probable reason for this higher antimicrobial effect of 3D printed films may again be the higher dispersion and organization of ZnONPs in the oriented film structure, causing a better interaction of ZnONPs with bacterial cells. It is a known phenomenon that better dispersion of nanoparticles in the nanocomposite films result in better antimicrobial activity (Antibacterial Activity of Polymer Nanocomposites Incorporating Graphene and Its Derivatives: A State of Art). This is due to the higher release rate of ions from the dispersed and unaggregated nanoparticles in the film matrix. ZnONPs, besides interacting with microbial cells and producing ROS, also cause antimicrobial effect via the release of Zn^2+^ ions, which are produced in higher amount when they are uniformly dispersed in the polymer. Although the the contribution of Zn^2+^ to the antimicrobial efficacy of ZnO particles is minor because too low concentrations of soluble zinc species are released from the dissolution of ZnO particles, it causes a synergistic effect combining with other phenomenon responsible for the antimicrobial activity of ZnONPs (The contribution of zinc ions to the antimicrobial activity of zinc oxide). Hence, the variation in dispersion of ZnONPs alters their release pattern, which may lead to a direct effect on antimicrobial activity.

### 3.6. Cytotoxicity and Cell Viability

Cytotoxicity of bio-inks is the paramount factor governing their practical applications in advanced practical research areas such as food and biomedical engineering. Therefore, the bio-inks should be non-cytotoxic, and the fabricated biopolymer materials must provide better interaction with biological surfaces such as cells for better adhesion, which is important for advanced applications such as tissue engineering and high-protein lab-grown foods. For the evaluation of cytotoxicity of the bio-ink formulations used for 3D printing, the WST-1 assay was used, and the results are presented in Figure 6. The neat Pec/CMC-3D bio-inks did not show any cytotoxicity towards NDFB cells, and the obtained cell growth was comparable to the control. Interestingly, the Pec/CMC/ZnONP-3D bio-inks were not only safe but also significantly (*p* < 0.0001) increased the viability of NDFB cells with their increased exposure concentration. The cell viability on exposure with 2% (w/v) of Pec/CMC/ZnONP-3D bio-inks was almost two folds higher compared with the control samples and the neat bio-inks, indicating that Pec/CMC/ZnONP-3D bio-inks induced fibroblast proliferation. ZnONP is well known for its role in angiogenesis, fibroblast proliferation, and as an active agent in wound dressings [42]. In addition, Ahmed et al. [43] reported 57.8% better healing effects of hydrogel patch incorporated with 0.5 M of ZnO on mouse skin wounds than control. These results strongly endorse the safety of 3D printed biomaterials and their potential application in various advanced areas of science and technology.

## 4. Conclusions

This study aimed to develop polymer blend and nanocomposite films by two different fabrication methods, i.e., solution casting and 3D printing, and compare their physicochemical and functional properties for practical application and potential commercialization. While the solution casting method is currently most prevalent for fabricating biopolymer films, it is not feasible for industrial-level scale-up. On the other hand, 3D printing is emerging as a more lucrative alternative process, especially for fabricating biopolymer-based food packaging films.

Here, ZnO nanoparticles were prepared and integrated with Pectin/CMC blend films. The formulations were cast into films using solution casting and 3D bioprinting methods. The films fabricated via the 3D bioprinting method had a more compact, ordered, and organized structure, which governed their physicochemical and functional properties. The 3D printed neat and nanocomposite films displayed better mechanical properties, hydrophobicity, moisture barrier, and antimicrobial activity than those fabricated using the solution casting method. The Pec/CMC/ZnO-3D composite films exhibited much stronger eradication of the *S. aureus* and *E. coli* O157: H7 bacterial strains within 6 h and 3 h of exposure, respectively, compared to 9 h and 6 h, respectively, for Pec/CMC/ZnO-SC films. Furthermore, the 3D printed nanocomposite films were not cytotoxic but displayed 2-fold increased fibroblast proliferation against NDFB cells. Therefore, the 3D printing technology exhibited tremendous potential as an alternative to the solution casting method for fabricating biopolymer-based films and scaffolds, with relatively better properties, for advanced applications such as biomedical engineering and food packaging. Further, it offers a potential for commercialization and presents a way in which biodegradable films could compete commercially with non-environmentally friendly petroleum-based plastics.

## Figures and Tables

**Figure 1 polymers-14-04711-f001:**
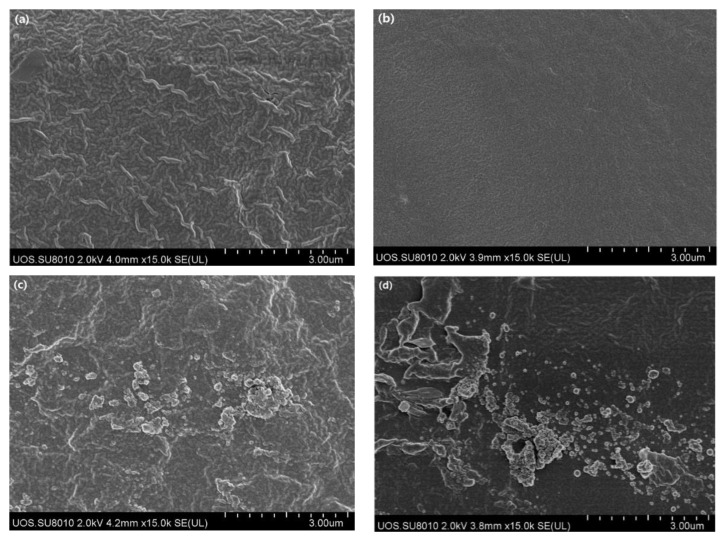
FE-SEM micrographs of the Pec/CMC-based bio-films. (**a**) Pec/CMC-SC, (**b**) Pec/CMC-3D, (**c**) Pec/CMC/ZnO-SC, and (**d**) Pec/CMC/ZnO-3D.

**Figure 2 polymers-14-04711-f002:**
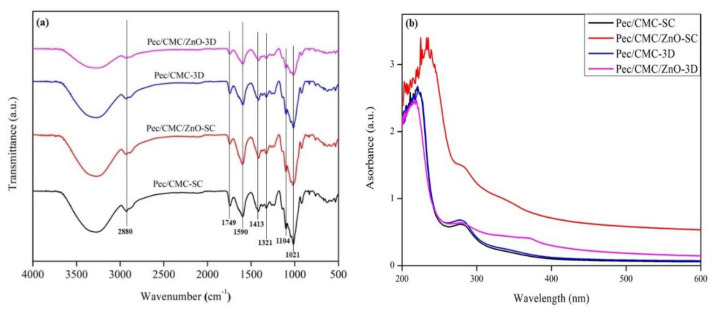
FT-IR (**a**) and UV-visible spectra (**b**) of the Pec/CMC based bio-films.

**Figure 3 polymers-14-04711-f003:**
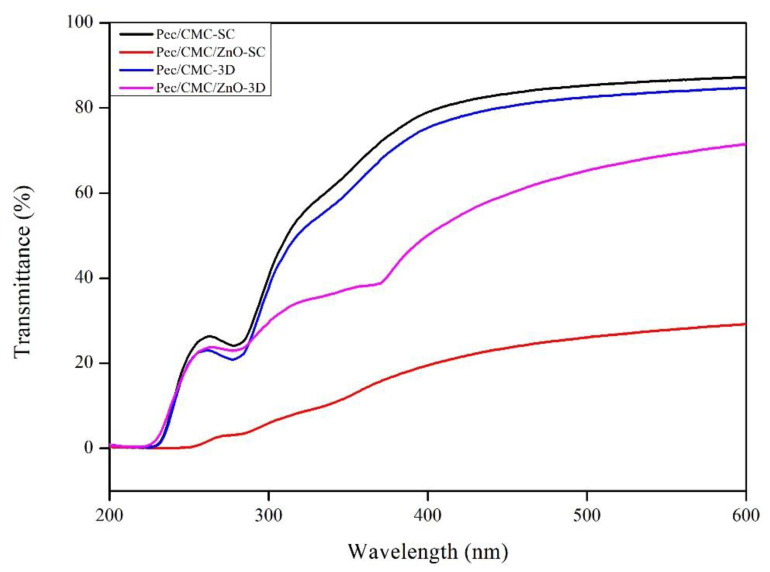
The light transmittance spectra of the Pec/CMC-based bio-films.

**Figure 4 polymers-14-04711-f004:**
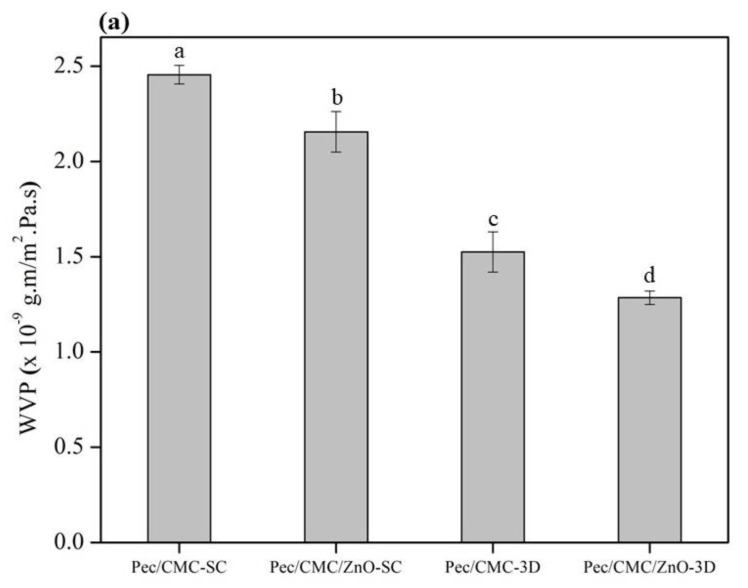
Water vapor permeability (**a**) and water contact angle (**b**) of the Pec/CMC based bio-films. a–d means in each bar with a different letter are significantly different by Duncan’s multiple range test at *p* < 0.05.

**Figure 5 polymers-14-04711-f005:**
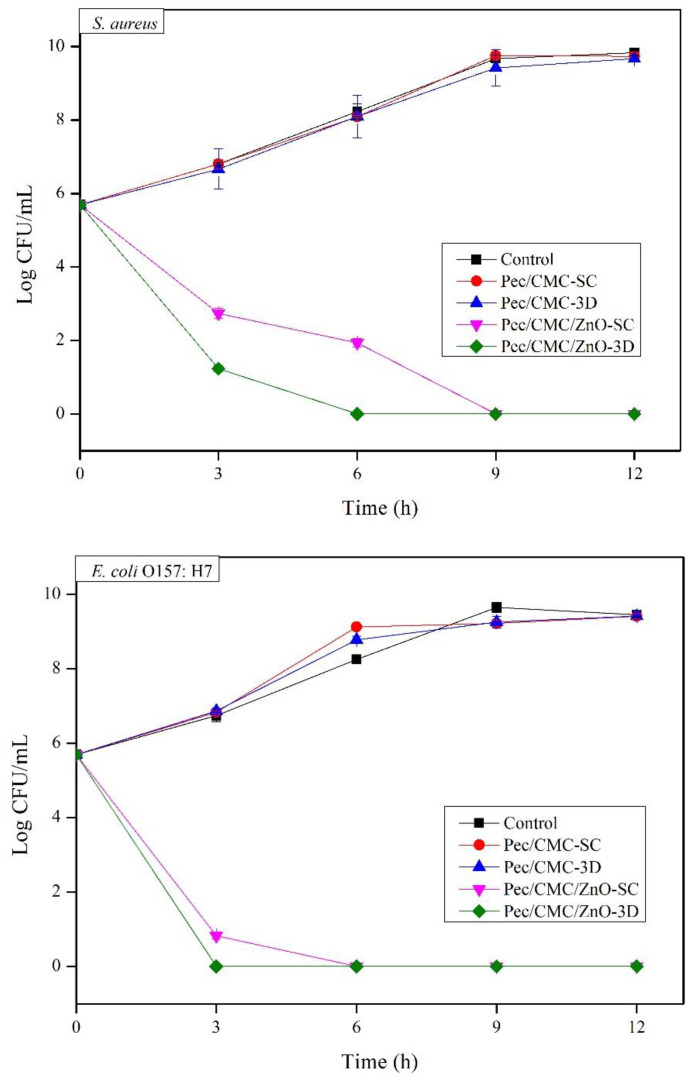
Antimicrobial activity of the Pec/CMC-based bio-films against *Staphylococcus aureus* and *E. coli* O157: H7 in media for 12 h.

**Figure 6 polymers-14-04711-f006:**
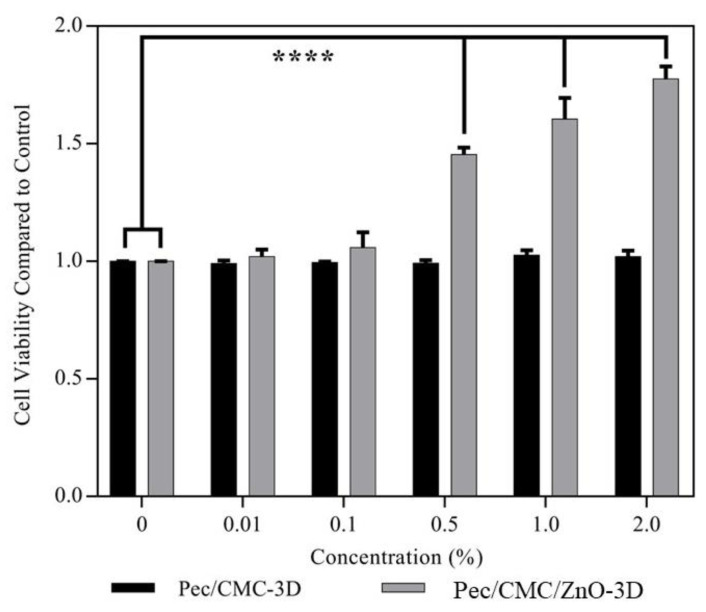
Cell viability of normal human dermal fibroblast (NDFB) cells treated with the Pec/CMC-based bio-inks at 0, 0.01, 0.1, 0.5, 1.0, and 2.0% (w/v). **** means values in each concentration are significantly different by ANOVA test at *p* < 0.0001.

**Table 1 polymers-14-04711-t001:** The formulation of Pec/CMC based bio-inks.

Components	Proportion (%)
Pectin	3.33%
Sodium carboxymethyl cellulose	1.67%
Glycerin	1.50%
Calcium chloride	0.11%
Water	93.39%

**Table 2 polymers-14-04711-t002:** The conditions of the 3D bioprinter for fabricating Pec/CMC-based films.

Parameter	Setting Value	Parameter	Setting Value
Nozzle size of syringe (Gauge)	25	Printing speed (mm/s)	10
Nozzle size (mm)	0.6	Move speed (mm/s)	2
Hight of layer (mm)	0.2	Retraction speed (mm/s)	10
Density (%)	100	Z HOP (mm)	0.4
Infill pattern	Concentric	Pressure (KPa)	150
The angle of infill rotation	90°	-	-
Bed temperature	Room temperature	-	-
Syringe temperature	Room temperature	-	-

**Table 3 polymers-14-04711-t003:** Surface color and mechanical properties of the Pec/CMC based bio-films.

Samples	*L**	*a**	*b**	∆*E*	TS (MPa)	EB (%)
Pec/CMC-SC	90.07 ± 0.05 ^b^	4.31 ± 0.01 ^b^	2.70 ± 0.02 ^c^	5.51 ± 0.02 ^c^	12.8 ± 0.6 ^a^	27.9 ± 3.2 ^c^
Pec/CMC/ZnO-SC	88.38 ± 0.19 ^d^	4.23 ± 0.01 ^c^	9.67 ± 0.33 ^a^	7.98 ± 0.31 ^a^	24.1 ± 1.5 ^b^	26.1 ± 2.6 ^c^
Pec/CMC-3D	89.67 ± 0.27 ^c^	4.36 ± 0.01 ^a^	2.39 ± 0.18 ^c^	5.83 ± 0.08 ^b^	27.6 ± 2.7 ^b^	19.2 ± 1.6 ^a^
Pec/CMC/ZnO-3D	90.56 ± 0.08 ^a^	3.72 ± 0.02 ^d^	4.12 ± 0.05 ^b^	4.48 ± 0.04 ^d^	41.3 ± 5.8 ^c^	22.1 ± 2.1 ^a^

The values are written as a mean ± standard deviation. Different letters within the same column indicated significant differences (*p* < 0.05).

## Data Availability

The data presented in this study are available on request from the corresponding author.

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
