# Peer review of "3D-Printed Pectin/Carboxymethyl Cellulose/ZnO Bio-Inks: Comparative Analysis with the Solution Casting Method"

_polymers, 2022, doi:10.3390/polym14214711_

Round 1

Reviewer 1 Report

The authors have provided a manuscript on the use of pectin/carboxymethyl cellulose/ZnO bio-inks in providing a functional films or scaffolds and biomedical applications.  I commend the authors on the work done and the putting together of an impressive body of work into a cogent paper.  I have a few questions and suggestions that I hope will enhance the reader appreciation of the work done.

1.       What is the main novel finding in your paper? 

2.       Could you please explain with reference on what is the motivation for you to choose pectin, CMC etc. for this paper.

3.       In conclusions, you are suggesting this as an alternative to the solution casting method for fabricating biopolymer-based films and scaffolds, could you explain why only few hours of Cytotoxicity and cell viability studies is enough to make this conclusion.

Author Response

Reviewer 1. The authors have provided a manuscript on the use of pectin/carboxymethyl cellulose/ZnO bio-inks in providing a functional films or scaffolds and biomedical applications. I commend the authors on the work done and the putting together of an impressive body of work into a cogent paper. I have a few questions and suggestions that I hope will enhance the reader appreciation of the work done.

Comment 1. What is the main novel finding in your paper?

Response: The main novel finding of this work is that the 3D bioprinting method can be an alternative, better, and more practical method for fabricating biopolymer films instead of the commonly used solution casting method which is difficult to scale-up on an industrial level. The novelty of this work has now been clearly highlighted included in the manuscript. (Page 1; line 26)

Comment 2. Could you please explain with reference on what is the motivation for you to choose pectin, CMC etc. for this paper.

Response: The motivation behind selecting pectin and CMC for this study has now been mentioned clearly in the manuscript. (Page 2; Lines 92-96)

Comment 3. In conclusions, you are suggesting this as an alternative to the solution casting method for fabricating biopolymer-based films and scaffolds, could you explain why only few hours of Cytotoxicity and cell viability studies is enough to make this conclusion.

Response: We appreciate the reviewer for their insightful comments. We tested the cytotoxicity for 24 h. In the previous studies of cytotoxicity, the results of cytotoxicity were not significantly different by treatment time between 24 h and 72 h.  Please see the following reference: https://doi.org/10.1177/026119299302100208 and Figure 1 in https://doi.org/10.3390/nano7070173. These studies tested cytotoxicity of media with treatment materials and scaffolds. Therefore, we tested cytotoxicity for 24 h as well as we decided no need data of 72 h because our results of cytotoxicity showed cell proliferation rather than cytotoxicity.

Reviewer 2 Report

The comment for Authors:

    The authors compared two types of pectin/CMC/ZnO films by using a casting method and a 3D printing technique. However, the quality of this paper is not suitable for the publication of Polymers now. It should be rejected by the following comments.

  1. In Figure 1. The authors showed only SEM images of two films and emphasized that the surface of printed films is smoother than casting films, and the ZnO particles could be oriented in the printed films. However, using an extrusion printer to fabricate the films usually have a fiber-stacked structure in the printed films, especially since the printing density is around 60% (table 2). Therefore, it lacks the overall low magnification images and cross-section of SEM images of the two types of films.
  2. In Figure 1 and FITR results. Again, the authors mentioned that the ZnO particles could be oriented in the films, and there are different degrees of H-bonding and free OH groups. However, there are not any results shown in the manuscript. For example, the authors should provide the SEM-EDX/XPS and TEM results to confirm their hypothesis.
  3. In 3.2. section, the authors mentioned that the two films have no different color and mechanical properties. It should provide images and mechanical diagrams of the films.
  4. In Figure 4. The authors show that pectin/CMC/ZnO printed films have the lowest WVP value. However, the 60% printed density of films usually still has a more porous structure among the printed fibers. Therefore, the author should provide the images of the printed films and describe more discussion to explain the more reasonable mechanism in this section.
  5. In Figure 5, the authors should provide the images of the S. aureus and E. coli cultured on the casting and printed films. Furthermore, the authors should add more discussion to explain why the printed film provides the better antimicrobial activity.

Author Response

Reviewer 2. The authors compared two types of pectin/CMC/ZnO films by using a casting method and a 3D printing technique. However, the quality of this paper is not suitable for the publication of Polymers now. It should be rejected by the following comments.

Comment 1. In Figure 1. The authors showed only SEM images of two films and emphasized that the surface of printed films is smoother than casting films, and the ZnO particles could be oriented in the printed films. However, using an extrusion printer to fabricate the films usually have a fiber-stacked structure in the printed films, especially since the printing density is around 60% (table 2). Therefore, it lacks the overall low magnification images and cross-section of SEM images of the two types of films.

Response: We appreciate the reviewer for their insightful comments. First of all, we apologize for the error, it was not 60% but 80% printing density. The typo error has now been removed. Also, for the hydrogels used here, the printing density was optimized between 60% and 100%. The morphology of the fabricated films was same within the printing density range 60-100%. The reason being a much higher spreadability of hydrogel materials as compared to commonly utilized polycaprolactone (PCL) or polylactic acid (PLA) polymers for 3D printing applications. In other words, while the PCL and PLA form fiber-like morphology at 60% printing density, hydrogels don’t form intact fibers and tend to spread and amalgamate to form a continuous structure. However, 60% printing density was the minimum density where the formation of continuous films were observed instead of fibers, 80% printing density was used for the current fabrication process.

Further, regarding the SEM images, there have been reports where the SEM images have been acquired at much lower magnifications, i.e., 400 μm scale bar. Still, the fiber stacking was visible. Please see the following reference: 10.3390/polym12061334. The magnigfication used in this study was still comparatively much higher, i.e. down to 3 μm scale bar. The reason for not finding this type of morphology even at 3 μm scale confirms the higher interaction and amalgamation of the stacked hydrogel layers. Hence, the stacked morphology is not seen in the SEM images. Besides, the nozzle size used for this fabrication is 0.6 mm or 600 μm (as mentioned in Table 2). Hence, the fiber size must be within the μm range. This again confirms that the amalgamation of individual fibers occurred resulting in the formation of a uniform film surface.  Therefore, as per our understanding, there was no requirement of taking images at even lower magnifications.

Comment 2. In Figure 1 and FITR results. Again, the authors mentioned that the ZnO particles could be oriented in the films, and there are different degrees of H-bonding and free OH groups. However, there are not any results shown in the manuscript. For example, the authors should provide the SEM-EDX/XPS and TEM results to confirm their hypothesis.

Response: We appreciate the reviewer’s concern. We want to clarify that orientation of ZnO was not mentioned in the manuscript. Instead, we discussed the better dispersion of ZnONPs in the polymer matrix due to the orientation of the biopolymeric formulation as a whole. The previous studies confirm that when 3D-printer works in a raster mode to form films, the material is directionally oriented. Please see the following reference: 10.3390/polym12061334. Since ZnONPs are present in the formulation, they will have a better dispersion in the films. Besides, the H-bonding between the biopolymers and ZnONPs is an established and proven phenomenon, and has also been reported previously by several researchers: 10.1016/j.ijbiomac.2016.09.110; 10.1007/s10965-014-0550-0. Please see the following reference published by our research group as well: 10.1016/j.susmat.2021.e00325. Apart from these, we conducted additional experiments to prove the H-bonding between ZnONPs and polymer (Fig. S1. Supporting data file).

Comment 3. In 3.2. section, the authors mentioned that the two films have no different color and mechanical properties. It should provide images and mechanical diagrams of the films.

Response: As suggested by the reviewer, the digital images of the films are provided in the Supporting data file. Further, regarding the mechanical properties, we didn’t mention that the mechanical properties are same for the films. Rather, we discussed with reasons, the higher mechanical properties of the 3D-printed films compared to the solution cast ones.

Regarding the mechanical property diagrams, we assume the reviewer is referring to the stress-strain curves. Unfortunately, it won’t be possible to provide the curves at this moment due to some technical issues. However, the values of tensile strength and elongation at break have been calculated and provided in Table 2, which indicate the properties of the films.

Comment 4. In Figure 4. The authors show that pectin/CMC/ZnO printed films have the lowest WVP value. However, the 60% printed density of films usually still has a more porous structure among the printed fibers. Therefore, the author should provide the images of the printed films and describe more discussion to explain the more reasonable mechanism in this section.

Response: We thank the reviewers for raising this query. We have already clarified the query regarding the print density and morphology of the fabricated material in response to Comment 1 above. An 80% print density was used and the hydrogel layers tend to amalgamate with each other to form a continuous structure. Hence, the morphology was not fibrous and porous but in the form of a continuous film.

Regarding the lower WVP of 3D-printed films, we want to clarify here that 3D-printing technique is a robust method to precisely control the morphology of the fabricated material. In this work, this technique was used to fabricate films, which are fabricated in raster mode, i.e., one row adjacent to the previous, and one layer on top of the other. This way, the material is deposited in perfect alignment and pores are not formed. Further, the absence of porous structure has already been confirmed by the SEM images obtained at a scale of 3 μm (Figure 1). The highly organized and ordered alignment of the material in the 3D-printed films results in compact structure, leading to lower WVP, which was also confirmed by the current study.

Comment 5. In Figure 5, the authors should provide the images of the S. aureus and E. coli cultured on the casting and printed films. Furthermore, the authors should add more discussion to explain why the printed film provides the better antimicrobial activity.

Response: I think we are not able to provide a clear explanation of the procedure followed which led the reviewer to misinterpret it. The bacterial cells were not cultured on the films. Rather, the culture was carried out in liquid broth of bacterial growth media to which the films were added. We have tried to improve the procedure (Page 5: lines 208 - 209) and discussed the results in more detail (Page 11: lines 414 - 426).

Reviewer 3 Report

The presented work describes the comparative study of the properties of the casted and 3D printed films composed of CMC/PEC and ZnO. The studies conducted are interesting; however, some additional information is required prior to the publication.

·         The Introduction is well written. However, lines 66-67 are too general - alginate does not solidify in lower temperatures. Please rewrite the sentence.

·         2.2. Preparation of zinc oxide nanoparticles – The authors indeed obtained zinc oxide particles, but no additional analysis was carried out. At this point you cannot assume that the material obtained is ZnO-NPs.

·         2.3. Preparation of Pectin/CMC/ZnO bio-inks and films – Why did you dilute the bioink intended for solution casting that way when you intended to compare the properties of both: printed and casted ones? It does not make any sense.

·         Table 2. What was the pressure used in the 3D printing? The % is not a proper pressure unit.

·         Lines 312-314 and so on: This refers to my first question: How can you compare the properties of the films, when you used lower concentration of the polymer for the casting (lower concentration=lower mechanical properties) and higher for the 3D printing? In my opinion those two should not be compared that way because the increase of the mechanical properties may be due to the use of 3D printing, but you cannot rule out the concentration factor.

Author Response

Reviewer 3. The presented work describes the comparative study of the properties of the casted and 3D printed films composed of CMC/PEC and ZnO. The studies conducted are interesting; however, some additional information is required prior to the publication.

Comment 1. The Introduction is well written. However, lines 66-67 are too general - alginate does not solidify in lower temperatures. Please rewrite the sentence.

Response: The sentence has been modified. (Page 2, Line 68)

Comment 2. 2.2. Preparation of zinc oxide nanoparticles – The authors indeed obtained zinc oxide particles, but no additional analysis was carried out. At this point you cannot assume that the material obtained is ZnO-NPs.

Response: The prepared zinc oxide nanoparticles used in this study were the same is used in our previously published works. The characterization data of the ZnONPs has already been published in detail. Please read the following article for more information: 10.1016/j.msec.2018.08.002, 10.1007/s10311-018-00835-z.

Comment 3. 2.3. Preparation of Pectin/CMC/ZnO bio-inks and films – Why did you dilute the bioink intended for solution casting that way when you intended to compare the properties of both: printed and casted ones? It does not make any sense.

Response: The formulations generally used for 3D bioprinters are prepared to have high viscosity, to make them suit the printing process. However, such viscous solution cannot be used for solution casting, as they are difficult to spread on the casting plates. Hence, the water content is generally kept higher in the solution casting formulations. This is where the whole difference in the properties creep in, which has been explained in detail in the manuscript. The higher water content in the solution cast films result in somewhat inferior physicochemical properties compared to the 3D-printed films with lower water content. Besides, this work is not comparing two different formulations, but two different technologies, their outcomes, and their suitability to be industrialized. The preparation of formulations is just a part of the technologies.

Comment 4. Table 2. What was the pressure used in the 3D printing? The % is not a proper pressure unit.

Response: We appreciate the reviewer’s question. We in put the wrong pressure unit. We revised the unit from % to MPa in Table 2.

Comment 5. Lines 312-314 and so on: This refers to my first question: How can you compare the properties of the films, when you used lower concentration of the polymer for the casting (lower concentration=lower mechanical properties) and higher for the 3D printing? In my opinion those two should not be compared that way because the increase of the mechanical properties may be due to the use of 3D printing, but you cannot rule out the concentration factor.

Response: We agree with the reviewer that the concentration factor can not be ruled out. In addition to the explanation provided for the reviewer’s first question, the effect of formulations on the mechanical properties has been indicated in the manuscript now. (Page 8, Line 324-325). The mechanism has been explained in Supporting data file.

Round 2

Reviewer 2 Report

The Authors reply all my comments

Reviewer 3 Report

I believe that this paper can be accpeted for publication.